# Development and Validation of a 7-eRNA Prognostic Signature for Lung Adenocarcinoma

**DOI:** 10.3390/biology14101431

**Published:** 2025-10-17

**Authors:** Yiwen Sun, Keng Chen, Jingkai Zhang, Zhijie Hu, Mingmei Xiong, Zhigang Fang, Guanmei Chen, Xiaomei Meng, Baolin Liao, Yuanyan Xiong, Luping Lin

**Affiliations:** 1Key Laboratory of Gene Engineering of the Ministry of Education, Department of Biochemistry, School of Life Sciences, Sun Yat-sen University, Guangzhou 510220, China; snswdqr@126.com (Y.S.);; 2Guangzhou Eighth People’s Hospital, Guangzhou Medical University, Guangzhou 510440, Chinapolinlbl@163.com (B.L.); 3Guangzhou Key Laboratory of Clinical Pathogen Research for Infectious Diseases, Guangzhou 510440, China; 4Guangdong Key Laboratory of Traditional Chinese Medicine for the Prevention and Treatment of Infectious Diseases, Guangzhou 510440, China

**Keywords:** drug prediction, prognosis evaluation model, LUAD, eRNA

## Abstract

**Simple Summary:**

Lung cancer, with the highest incidence and mortality rates among malignant tumors in recent years, saw approximately 2.48 million new cases globally in 2022, accounting for 12.4% of all new cancer cases. Lung adenocarcinoma is the most prevalent subtype of lung cancer, representing about 40% of all lung cancer cases. Identifying biomarkers for prognostic assessment of lung adenocarcinoma to improve patients’ 5-year survival rate and prognosis is of utmost urgency. With the advancement of bioinformatics technologies, measuring the expression levels of enhancer RNAs has become increasingly convenient and stable. Enhancer RNAs play crucial roles in disease progression. However, research on the functions of enhancer RNAs in lung adenocarcinoma is still insufficient. In this study, data from the databases were used to construct a prognostic assessment model for lung adenocarcinoma based on enhancer RNAs. A 7-enhancer RNA model was developed and validated for its effectiveness and robustness in both the train and test sets. Using this 7-enhancer RNA model, samples were divided into high-risk and low-risk groups. Enrichment pathways, tumor-immune microenvironments, drug sensitivity analysis and somatic mutations were analyzed for these two groups. These results are expected to promote in-depth explorations of the mechanisms underlying lung adenocarcinoma development and open new perspectives for enhancer RNA regulation mechanism research.

**Abstract:**

Enhancer RNAs (eRNAs) are abundant in most human cells and tissues, and quantifying eRNAs has become a robust approach for biomarker discovery. While eRNAs play crucial roles in regulating biological processes and cancer progression, their functions in lung adenocarcinoma (LUAD) remain poorly understood. Here, we developed a LUAD prognostic model based on eRNA expression data from The Cancer Genome Atlas (TCGA). Through rigorous validation, a 7-eRNA signature was identified, which robustly stratified LUAD patients into high-risk and low-risk groups in both training and testing sets. Functional analyses revealed distinct enrichment of pathways related to amino acid biosynthesis, ribosome biogenesis, and proteasome activity in high-risk patients. Somatic mutation profiling highlighted TP53 and TTN as frequently mutated genes, while drug sensitivity prediction identified four potential therapeutic agents (including AZD4547 and Nutlin-3a) for high-risk individuals. Collectively, this study constructed a 7-eRNA prognostic model for LUAD, providing a powerful tool for clinical risk assessment and uncovering eRNA-mediated regulatory mechanisms.

## 1. Introduction

Lung adenocarcinoma (LUAD) is the most prevalent subtypes of non-small cell lung cancer (NSCLC), accounting for approximately 40% of all lung cancer cases. Globally, there were approximately 2.48 million newly diagnosed lung cancer cases in 2022, representing 12.4% of all new cancer diagnoses [1]. The primary therapeutic modalities for NSCLC include surgery, radiotherapy and chemotherapy [2]. However, only a small proportion of patients with early-stage NSCLC are eligible for surgical intervention. Notably, the 5-year survival rate for patients with stage I NSCLC can reach as high as 58.4% [3]. In contrast, patients with advanced NSCLC such as stage ⅢB and ⅢC typically lack surgical indications, and their 5-year survival rate drops to about 26% and13%, indicating a poor prognosis [4]. Therefore, early diagnosis of lung cancer is critical, and identifying reliable biomarkers for prognosis assessment in NSCLC could significantly improve patient 5-year survival rates.

Prognostic biomarkers for lung cancer are critical biological indicators that enable the prediction of survival duration, recurrence risk, disease progression, and treatment response in lung cancer patients [5]. Although there have been advancements in research on such biomarkers, their clinical application remains hampered by several limitations and challenges. Firstly, the sensitivity and specificity of many biomarkers are yet to reach optimal level, increasing the risk of false-positive and false-negative results [6]. Secondly, the stability and reproducibility of these biomarkers represent significant concerns, undermining their reliability in clinical settings [7]. Thirdly, the economic costs associated with the clinical application of biomarkers for prognostic assessment are non-negligible, restricting their widespread adoption [8]. Therefore, further efforts are required to advance the discovery, validation, standardization, and clinical translation of lung cancer prognostic biomarkers.

Enhancer RNAs (eRNAs) are a class of non-coding RNAs transcribed from genomic enhancer regions. Enhancers exert critical regulatory functions roles across hundreds of different cell types, and their dysregulation emerged as a driving force in the pathogenesis of numerous human cancers [9]. In human cancer tissues, the differential expression patterns of eRNAs highlight their potential as therapeutic targets and biomarkers [10]. Functionally, eRNA can influence tumor growth by regulating anti-oncogenes expression [11]. Moreover, they are implicated in the modulation of cancer-related signaling pathways, where they play pivotal roles in mediating target gene activation through transcriptional circuitry [12]. Mechanistically, eRNAs promote transcription by multiple modalities: regulating chromatin accessibility at target gene promoters, binding directly to these promoters [13,14], or forming eRNA-protein complexes that facilitate enhancer–promoter looping [15]. Accumulating studies have illuminated the roles of eRNA in cancer. For example, in breast cancer, activation of Estrogen Receptor 1 (ESR1) broadly induces increased eRNA transcription [16], indicating the association between eRNAs and the activation of oncogenes or oncogenic signaling pathways.

Currently, in-depth investigations into the biological characteristics of lung cancer have identified a series of prognostic markers. As key regulators of gene expression and tumorigenesis, eRNAs hold significant functional importance. However, research exploring the roles and mechanisms of eRNAs in lung cancer remains limited. Accordingly, this study aims to establish a prognostic model for LUAD using eRNAs as indicators, followed by comprehensive analyses to validate its accuracy and reliability. Specifically, we developed a robust prognostic model suitable for clinical application, validated its performance, and conducted relevant analyses of gene enrichment pathways and potential therapeutic drugs in LUAD.

## 2. Materials and Methods

### 2.1. Data Collection and Preprocessing

RNA-seq count data, eRNA expression profiles (RPKM), and clinical information for LUAD were retrieved from The Cancer Genome Atlas (TCGA). Specifically: RNA-seq count data included 586 samples (527 tumor, 59 normal); eRNA expression data contained 559 tumor samples, with an expression matrix of 302,951 eRNAs across 500 tumor and 59 normal samples; Clinical data were available for 499 LUAD patients with complete records. A merged dataset of 499 samples was generated by intersecting RNA-seq data, eRNA profiles, and clinical information. Subsequent analyses utilized four datasets: RNA-seq counts, eRNA expression (including 59 normal samples), and clinical data for the 499 patients.

Clinical data preprocessing included: Survival time integration: “days to death” (for deceased patients) or “days to last follow-up” (for living patients) were unified as survival days; Tumor staging: consolidated into four categories (Stage I–IV) without subclassification; Retained variables: survival days, survival status, age, tumor stage, and TNM (Tumor, Node, Metastasis) stage (6 variables total). Categorical variables were converted to numerical codes for computational compatibility (Table 1).

In the TNM staging system, “T” stands for “Tumor” and describes the extent of the primary tumor. As tumor size increases and local tissue involvement grows, the designation progresses from T1 to T4. “N” stands for “Node” and indicates the involvement of regional lymph nodes. When no regional lymph nodes are affected, it is labeled N0; increasing degrees of nodal involvement are denoted by N1 to N3. “M” stands for “metastasis” and refers to distant (hematogenous) spread. Patients without distant metastases are classified as M0, those with distant metastases as M1, and MX indicates that distant metastasis cannot be assessed. Based on these three parameters, specific stages are assigned by combining the T, N, and M categories.

### 2.2. Differential Expression Analysis of eRNAs

A total of 302,951 eRNAs were initially identified across all samples. After filtering out low-expression data, 66,976 eRNAs were retained for further analysis. Differential expression analysis between tumor and normal samples was performed using the “limma” package [17] in R (version 4.4.2), with thresholds set as |log_2_FC| > 1 and FDR < 0.001. This yielded 6280 differentially expressed eRNAs. The analysis followed the core workflow of limma, including linear regression and differential calculation steps such as voom, fit, and eBayes. Output results were sorted by *p*-value to determine the significance of differential expression for each eRNA. The Sangerbox 3.0 analysis tool was utilized to support this process [18].

### 2.3. Establishing and Evaluating Prognostic 7-eRNA Model

First, univariate Cox proportional hazards analysis was performed to evaluate 6280 differentially expressed eRNAs. The LASSO method was then used to further refine the eRNA candidates [19], and 10-fold cross-validation was used during LASSO regression. Subsequently, multivariate Cox proportional hazards analysis was applied to filter the remaining variables, resulting in the establishment of a 7-eRNA model. For each individual, the risk score was calculated by multiplying the regression coefficients of the 7 eRNAs by their respective expression levels. Using the median risk score as a cutoff, all samples were divided into two subgroups. Kaplan–Meier (KM) curves and receiver operating characteristic (ROC) curves were used to assess the model’s performance [20]. Finally, the model was validated in both test cohorts and the combined cohort to verify its universality, with a two-sided log-rank test (*p* < 0.05) used for statistical significance.

To further test the model’s robustness, samples with incomplete TNM staging data were excluded, leaving 148 patients in the test group, 351 in the training group, and 499 in the combined group. Multivariate Cox regression analysis was performed to evaluate the association between risk scores and clinical variables, revealing that risk scores were statistically significant in the training, test, and combined groups—confirming the model’s robustness. ROC curve analysis, conducted using the “survivalROC” R package (version 1.0.3.1), demonstrated that the model outperformed other clinical features in predictive performance. Additionally, to facilitate the prediction of 1-, 2-, and 3-year overall survival (OS) probabilities in LUAD patients, nomograms were constructed using the “rms” R package.

### 2.4. Gene Set Enrichment Analysis (GSEA)

The model divided 499 LUAD patients into two subgroups. We conducted differential expression analysis between these two groups, screening differentially expressed genes using thresholds of |log_2_FC| ≥ 0.4 and FDR < 0.05. These genes were then subjected to GSEA, KEGG pathway enrichment analysis was conducted using the “clusterProfiler” R package, and results revealed the top four enriched pathways in the high-risk group.

### 2.5. Drug Sensitivity Analysis

Drug sensitivity prediction was performed using the “oncoPredict” R package to identify potential therapeutic agents for high-risk LUAD patients. This tool predicts in vivo drug responses and biomarkers from cell line screening data by training models on known expression matrices and drug sensitivity profiles [21]. In this study, gene expression and IC50 data from the Genomics of Drug Sensitivity in Cancer (GDSCv2) database—with 10-fold cross-validation as the default—were used as the training set. The “oncoPredict” package (version 1.2) was applied for data preprocessing, feature selection, model training, and evaluation. Drug sensitivity in LUAD patients was then predicted using their differentially expressed gene matrices, with significance assessed via the Mann–Whitney test.

### 2.6. Tumor Immune Microenvironment and Somatic Mutation Analysis

The “estimate” R package was used to predict stromal and immune cell infiltration in tumors via single-sample gene set enrichment analysis (ssGSEA), calculating stromal score, immune score, and ESTIMATE score. The “maftools” R package was employed for the analysis and visualization of cancer somatic mutation data.

### 2.7. eRNA-Gene Interaction Analysis

For the 7 eRNAs included in the prognostic model: Differential expression was analyzed between the control and experimental groups. Interactions between these eRNAs and genes (from TCGA) were visualized using Sankey diagrams. Functional annotations of the associated genes were summarized.

## 3. Results

### 3.1. Differentially Expressed of 6280 eRNAs in LUAD

Using expression data of 302,951 eRNAs from 499 LUAD patients and 59 normal samples (TCGA), we identified differentially expressed eRNAs through stringent filtering and statistical analysis. Firstly, the expression matrix was filtered to retain eRNAs expressed in ≥75% of the samples, resulting in 66,976 eRNAs. Differential expression analysis using the limma R package (FDR < 0.001, |log_2_FC| > 1) identified 6280 differentially expressed eRNAs in LUAD, including 2588 upregulated and 3692 downregulated eRNAs (Figure 1A). The heatmap (top 50 eRNAs) showed clustered expression profiles between tumor and normal samples (Figure 1B). These analyses provided a comprehensive overview of eRNA dysregulation in LUAD, prioritizing candidates for downstream functional and prognostic modeling.

### 3.2. Construction and Validation of a 7-eRNA Prognostic Model for LUAD

We performed stratified sampling (3:7 ratio) on 499 LUAD patients to generate a training set (351 samples) and testing set (148 samples). Using 6280 differentially expressed eRNAs from the training set, we constructed a survival prognostic model via sequential statistical analyses. Univariate Cox regression identified 16 eRNAs significantly associated with overall survival (OS, *p* < 0.001), which were further refined by LASSO regression to 10 eRNAs (Figure 2A,B). We subjected the 10 candidate eRNAs to multivariate Cox regression analysis, ultimately selecting 7 for the final prognostic model (chr1:206282970, chr2:200325425, chr5:131823242, chr11:126211724, chr15:41152185, chr17:3630257, chr22:42671972), including 6 risk factors and 1 protective factor (Figure 2C). The prognostic eRNAs and their corresponding parameters obtained from multivariable Cox regression analysis were shown below (Table 2).

The 7-eRNA risk score formula was derived from regression coefficients: risk scores = 0.04577 × chr1 + 0.20554 × chr2−1.11341 × chr5 + 0.02812 × chr11 + 0.05766 × chr15 + 0.22138 × chr17 + 0.24401 × chr22. Patients were stratified into high-risk (*n* = 176) and low-risk (*n* = 175) groups based on the median risk score. The risk scores exhibited an inverse association with patient survival in the LUAD patients (Figure 2D). Survival analysis showed a significant difference in OS between groups (Log-rank *p* < 0.001), with 2000-day survival rates of 25% (high-risk) versus 65% (low-risk) (Figure 2E). This indicated that high-risk patients exhibited significantly poorer prognosis compared to low-risk counterparts. The predictive accuracy of the risk score was evaluated using the Area Under the Curve (AUC), yielding an AUC of 0.726 in the training set (Figure 2F). This indicates the risk score effectively predicts patient prognosis, and further validated the model’s ability to distinguish survival outcomes.

### 3.3. External Validation and Independent Prognostic Value of the 7-eRNA Model in LUAD

To assess the generalizability of the 7-eRNA model, we applied it to two validation cohorts: a testing set (*n* = 148) and a combined dataset (*n* = 499). In the testing set, patients were stratified into high-risk (*n* = 74) and low-risk (*n* = 74) groups based on the median risk score. ROC curve analysis yielded an AUC of 0.634 (Figure 3A), confirming reliable predictive performance. Similarly, application to the combined dataset (*n* = 499) resulted in high-risk (*n* = 249) and low-risk (*n* = 250) groups with an AUC of 0.692 (Figure 3A). In both the testing set and combined dataset, the risk score was inversely associated with patient survival (Figure 3B). Heatmap visualization revealed the distribution of modeled eRNAs and clinicopathological variable. As the risk scores increases, the survival status of patients becomes worse and the proportion of deceased patients increases. The expression levels of chr5:131823242 which is protective factor were higher in patients with lower risk scores, and the expression levels of the other 6 eRNAs were higher in patients with higher risk scores. Their expression tendencies are consistent with the 6 protective factors and 1 risk factor found in our 7-eRNA model (Figure 3B). Collectively, these results demonstrate that the 7-eRNA model effectively discriminates survival outcomes across independent LUAD cohorts, highlighting its robustness and potential for clinical translation. To validate the robustness of the 7-eRNA model, we integrated clinical parameters (age, pathological stage, and TNM stage) with risk scores derived from the model. Univariate and multivariate Cox regression analyses were performed in the training set (*n* = 351), testing set (*n* = 148), and combined set (*n* = 499). Univariate analysis identified pathological stage, T stage, N stage, and risk score as significant predictors of OS in all cohorts (Figure 3C). Multivariate Cox regression further confirmed the 7-eRNA model as an independent prognostic factor in both the training set (HR = 1.237, 95% CI = 1.160–1.319, *p* < 0.001; Figure 3D) and combined dataset (HR = 1.068, 95% CI = 1.034–1.103, *p* < 0.001; Figure 3D). These results demonstrate that the 7-eRNA model retains prognostic value independent of traditional clinical variables, enhancing its utility for LUAD risk stratification.

### 3.4. Clinical Nomogram Integrating 7-eRNA Risk Score and Clinical Factors Enhances LUAD Prognosis Prediction

Clinical nomograms were constructed using data from 499 LUAD patients to integrate the 7-eRNA risk score with traditional clinical factors (age, pathological stage, and TNM stage). The nomograms predicted 1-, 2-, and 3-year survival probabilities and were validated in train (Figure 4A), test (Figure 4B), and combined (Figure 4C) sets. ROC curve analysis showed the risk score achieved the highest AUC among all variables in the train set (AUC = 0.73; Figure 4D), test set (AUC = 0.63; Figure 4E) and combined dataset (AUC = 0.69; Figure 4F). Notably, the risk score outperformed TNM stage in all cohorts, despite TNM being a gold-standard prognostic indicator. These results highlight the 7-eRNA model’s potential as a robust and complementary tool for LUAD prognosis.

### 3.5. Pathway Enrichment in High-Risk LUAD Patients

To characterize functional differences between risk groups, we performed GSEA on differentially expressed genes (DEGs) in high-risk vs. low-risk LUAD patients (Figure 5A). Using |log_2_FC| ≥ 0.4 and FDR < 0.05 as thresholds, 97 DEGs were identified. Visualization of the data showed the top 4 pathways enriched in high-risk patients: amino acids biosynthesis, DNA replication, eukaryotic ribosome biogenesis, and proteasome signaling (Figure 5B). Eukaryotic ribosome biogenesis was associated with increased production of rRNAs and ribosomal proteins, while amino acid biosynthesis linked to central carbon metabolism and key enzymes (e.g., aminotransferases). These pathways may drive tumor growth by enhancing protein synthesis and metabolic reprogramming. Modulation of eRNA expression likely contributes to pathway dysregulation, providing mechanistic insights into risk group heterogeneity.

### 3.6. Potential Therapeutic Agents for High-Risk LUAD Patients

Given the challenges of traditional drug development (long timelines and high costs), we leveraged the “oncoPredict” R package to identify repurposable drugs with differential sensitivity in high-risk patients. Using gene expression data and drug half-maximal inhibitory concentration (IC_50_) values from the GDSC project (198 drugs across 805 cell lines), we developed predictive models to assess chemosensitivity in TCGA LUAD cohorts. Differential expression profiles from high-risk vs. low-risk groups were inputted, and drug sensitivity was evaluated via estimated IC_50_ values. Four drugs showed significant sensitivity differences (Mann–Whitney test, *p* ≤ 0.05) and were prioritized for high-risk patients (Figure 6A–D). (1) AZD4547_1786: An FGFR antagonist that inhibits fibroblast growth factor receptor signaling, linked to tumor angiogenesis and proliferation [22]. (2) Nutlin−3a (-)_1047: A selective MDM2 inhibitor that stabilizes p53, inducing apoptosis in p53-wildtype tumors [23]. (3) PRT062607_1631: A Syk kinase inhibitor that disrupts B-cell receptor signaling, potentially reducing immune evasion in LUAD [24]. (4) WZ4003_1614: A dual NUAK/EGFR inhibitor targeting EGFR T790M-mutant NSCLC, with activity against both wild-type and mutant EGFR [25,26]. EGFR mutations (e.g., exon 19 deletions, L858R) are prevalent in LUAD and drive constitutive activation of EGFR tyrosine kinase, promoting tumor proliferation and metastasis via PI3K-AKT [27] and MAPK-ERK pathways [28], which regulate cell cycle progression [29,30]. We hypothesize that eRNAs modulate gene expression to influence cell cycle-related signaling, thereby affecting drug sensitivity. The identified drugs may exert efficacy by inhibiting these pathways, offering opportunities for personalized therapy in high-risk LUAD.

### 3.7. Somatic Mutation Differences Distinguish High- and Low-Risk LUAD Groups

The tumor immune microenvironment (TIME) contributes to LUAD heterogeneity and influences treatment response. To investigate its role in our prognostic model, we evaluated stromal scores, immune scores, and ESTIMATE scores using the “estimate” R package [31]. Violin and box plots revealed no significant differences in immune or stromal cell components between high-risk and low-risk groups (Appendix A). These results suggest TIME does not drive prognostic disparities between risk groups, indicating that factors like immune cell infiltration and stromal composition are not key determinants of survival differences in this cohort.

To characterize genetic differences between high-risk and low-risk patients, we performed somatic mutation analysis using TCGA LUAD data. Mutation profiles revealed that over 80% of samples in both groups carried somatic mutations (Figure 7A,B), highlighting the pervasive role of genetic alterations in LUAD. The top five most frequently mutated genes in both groups were TP53, TTN, CSMD3, MUC16, and RYR2. TP53, a tumor suppressor encoding a 53 kDa transcription factor, regulates G1/S cell cycle arrest and apoptosis in response to DNA damage. Loss of TP53 function leads to uncontrolled cell proliferation and genomic instability, with mutations present in >50% of malignancies [32]. TTN encodes titin, a sarcomeric protein critical for muscle mechanics. TTN mutations (particularly missense variants) correlate with improved chemosensitivity and OS, and combined TTN/TP53 mutations are associated with better treatment responses in LUAD [33]. Chi-square analysis of the top 20 mutated genes revealed significant intergroup differences in mutation frequencies (*p* < 0.001; Figure 7C), validating the model’s ability to stratify patients based on genetic landscapes. While the mutational landscapes of high-risk and low-risk groups shared common driver genes, intra-gene mutation types (e.g., missense, nonsense) differed significantly, suggesting potential for mutation-based therapeutic stratification.

### 3.8. Differential Expression and Functional Annotations of eRNAs in the 7-eRNA Model for LUAD

To characterize the biological relevance of the 7-eRNA model, we compared eRNA expression between 499 LUAD tumors and 59 normal samples. All seven eRNAs exhibited significantly differential expression (tumor vs. normal, *p* < 0.0001; Figure 8A). Using TCGA-derived enhancer-gene correspondence data, we identified potential target genes within 1MB genomic intervals for six of the seven eRNAs (Figure 8B), as only one eRNA had a pre-annotated gene association.

Three eRNAs with prominent risk coefficients (chr1:206282970, chr5:131823242, chr15:41152185) were prioritized for functional analysis: (1) chr1:206282970 (CTSE): Encodes cathepsin E, an aspartic protease overexpressed in LUAD. CTSE promotes tumor cell apoptosis by releasing tumor necrosis factor-related apoptosis-inducing ligand (TRAIL) and enhances chemosensitivity to doxorubicin in prostate cancer models [34,35,36]. Its proteolytic activity serves as a potential biomarker for cancer diagnosis and targeted therapy [37,38]. (2) chr5:131823242 (C5orf56): A long non-coding RNA located in a cytokine cluster (IL3/IL4/IL5) associated with type II inflammation. C5orf56 regulates airway remodeling in asthma/COPD and maintains gut homeostasis in inflammatory bowel disease (IBD) [39,40], suggesting a role in tumor-immune interactions. (3) chr15:41152185 (SRP14/EIF2AK4): The gene corresponding is SRP14 and EIF2AK4. (3-1) SRP14: Forms a heterodimer with SRP9 to regulate co-translational protein translocation and stress granule dynamics, linking to ribosome biogenesis pathway enrichment in high-risk LUAD [41,42]. (3-2) EIF2AK4: Phosphorylates eIF2α to inhibit protein synthesis under amino acid starvation, with mutations linked to pulmonary veno-occlusive disease (PVOD) [43]. Collectively, these findings suggest C5orf56 and SRP14/EIF2AK4 may influence LUAD progression through inflammation-immune crosstalk and translational stress pathways, respectively.

## 4. Discussion

Prognostic biomarkers are critical indicators for predicting survival duration, recurrence risk, disease progression, and treatment response, with significant implications for therapeutic decision-making. For instance, serum tumor markers such as carcinoembryonic antigen (CEA) [44], cytokeratin 19 fragment (CYFRA 21-1) [45], and neuron-specific enolase (NSE) [46] reflect disease severity and treatment outcomes through concentration fluctuations. Genetic alterations in EGFR [47], ALK [48], and ROS1 [49] correlate with favorable responses to targeted inhibitors and improved prognosis. PD-L1 expression levels predict immunotherapy efficacy, with high expressors tending to achieve better outcomes [50,51]. Additionally, the TNM staging system (incorporating tumor size, lymph node metastasis, and distant spread), histological grading, circulating tumor DNA (ctDNA), and circulating tumor cells (CTCs) serve as complementary prognostic tools. Emerging molecular markers, including miRNAs, proteomic signatures, and epigenetic modifications, are also under investigation as potential prognostic indicators [52,53,54]. Despite advancements, clinical application of lung cancer biomarkers faces challenges: suboptimal sensitivity and specificity of many markers, leading to false results [6]; poor stability and reproducibility across detection platforms or time points [7]; tumor heterogeneity and individual variability, limiting the utility of single markers [55]; dynamic tumor evolution, complicating real-time monitoring; high detection costs, hindering widespread adoption [8]; and the need for robust bioinformatics tools to integrate large-scale datasets into clinical decisions.

Addressing these limitations requires efforts to advance biomarker discovery, validation, standardization, and clinical translation—including developing cost-effective detection technologies and strengthening interdisciplinary collaboration across clinical, biological, and computational fields. Novel prognostic biomarkers and LUAD-specific models may provide invaluable insights into disease pathogenesis and therapeutic strategies. Technological advancements (e.g., CAGE-seq, RNA-seq) and eRNA databases (HeRA [56], eRic [57], and TCGA) have enabled eRNAs to emerge as promising prognostic biomarkers, with prior studies validating their utility in head and neck squamous cell carcinoma [58], hepatocellular carcinoma [59], and thyroid cancer [60]. Our study developed a unique 7-eRNA prognostic signature and validated its efficacy, offering new perspectives on LUAD regulatory mechanisms.

The GSEA of the high-risk population obtained from the prognostic model constructed in this study mainly focuses on the following pathways: biosynthesis of amino acids, ribosome biogenesis in eukaryotes, and the proteasome. In the drug sensitivity analysis of the high-risk population, potential drugs are mainly FGFR antagonists, MDM2 inhibitors, Syk kinase inhibitors, and NUAK kinase inhibitors, which are also EGFR inhibitors. In the immune microenvironment analysis, no significant differences were found in the immune microenvironment between the high-risk and low-risk populations, indicating that TIME factors, such as immune cell infiltration, cytokine levels, and stromal cells, may not be key prognostic determinants in distinguishing between high-risk and low-risk patients, and more in-depth research might be needed to fully understand their relationship with prognosis. In the somatic mutation analysis, there were significant differences in gene mutation types between the two subgroups, with higher mutation frequencies of TP53, TNN, and other genes in the high-risk group.

Furthermore, our study explored the genes corresponding to the seven eRNAs in LUAD, and investigated and summarized the functions of the genes corresponding to the three eRNAs with relatively significant risk values, namely chr1:206282970, chr5:131823242, and chr15:41152185. The gene regulated by chr1:206282970 is CTSE, which encodes cathepsin E and has been identified as a promising biomarker for multiple diseases. The expression levels and proteolytic activity of CTSE can be used for early detection and diagnosis of various malignancies and have shown promising results in the development of cancer-targeted chemotherapeutic drugs [34]. The gene regulated by chr5:131823242 is C5orf56, whose transcript is a kind of lncRNA and has been reported to be associated with asthma and COPD [39]. The genes regulated by chr15:41152185 are SRP14 and EIF2AK4. SRP14 is a key component of the SRP and is involved in co-translational protein translocation. Its main functions include translation regulation, stress regulation, and Alu RNA synthesis [41,42]. EIF2AK4 encodes eukaryotic translation initiation factor 2α kinase 4, which is involved in translation initiation and protein synthesis. It is also involved in various stress responses and is associated with PVOD, suggesting its important role in the development and prognosis of lung diseases, including lung cancer [43]. The above studies have identified LUAD-related genes, providing guidance for future research into the pathogenesis and treatment of LUAD, and these genes can be used as research targets for LUAD-related studies.

The 7-eRNA model benefits from technical advancements: CAGE-seq and eRNA-seq have reduced measurement costs and improved accuracy, facilitating clinical translation. Databases like HeRA, eRic, and TCeA support extending this approach to other cancers. Additionally, eRNA’s tissue specificity enhances prognostic precision and utility in mechanistic studies and targeted therapy development.

Although eRNAs provided new insights for LUAD’s research, our work has several limitations. Firstly, the structure of eRNAs is not clear, because there is a lack of information on remote chromosomal interactions, such as high-throughput chromosomal conformation capture (Hi-C) [61]. Secondly, the sample size used in our study was insufficient, which affected the efficiency of statistical analysis. Finally, although the accuracy and stability of the 7-eRNA model were demonstrated, we recognized some limitations.

Our model achieved good ROC performance in the training set (AUC = 0.726, Figure 2F) and the combined set (AUC = 0.692, Figure 3B), whereas its performance in the test set was modest (AUC = 0.646, Figure 3A), most likely due to the limited sample size of the test cohort (*n* = 148). Importantly, in multivariate Cox analysis (Figure 3D) the riskScore remained an independent prognostic factor (*p* < 0.001). Taken together with the ROC comparisons (Figure 4D–F), our model outperforms TNM staging and retains robust and clinically meaningful predictive power.

In our research, the HRs of riskScore are not as high as ‘Stage’, and riskScore has statistically insignificant hazard ratios in the test set (Figure 3C,D). These can be explained as follows: Univariate Cox regression analyses were first performed to assess the prognostic impact of individual variables. As shown in Figure 3C, ‘T’ group, stage, and the riskScore were all significantly associated with overall survival (OS). Multivariate Cox regression analyses in both the training set and the combined set further confirmed that the 7-eRNA model is a robust and independent predictor of OS (*p* < 0.001 in both cohorts). Although its HR was lower than that of Stage, the 7-eRNA model still provides valuable prognostic information and can therefore be used alongside Stage to refine clinical decision-making.

Stage is derived from the well-established TNM staging system that has been widely applied for decades, whereas our prognostic model is built on eRNAs and focuses more narrowly on the molecular regulatory mechanisms influencing patient outcomes, which might explain why Stage exhibits a higher HR in our analyses. Additionally, the relatively small sample size available for this study might limit the model’s stability, resulting in a *p*-value > 0.01 when the model is applied to the test set which has smaller sample size (*n* = 148). Above all, the 7-eRNA model retains independent prognostic value in the combined set (*p* < 0.001) and enhances prediction when integrated with clinical variables (via nomogram).

To assess the model’s accuracy and robustness with ROC curves, we observed that the AUCs of M were all below 0.5 in the train, test and combined sets (Figure 4D–F). This can be explained as follows: In routine clinical practice the final overall stage is assigned only after T, N and M have all been determined; consequently, evaluating M alone is less accurate and carries less prognostic information than evaluation of the composite stage. We therefore believe that Stage is a more representative and accurate predictor of disease outcome than any single component (T, N or M), and previous study [62] has likewise used overall Stage as the prognostic benchmark.

Furthermore, the eRNA approach offers three distinct advantages compared with other existing prognostic signatures. Firstly, eRNAs are closer to the functional source. eRNAs are transcribed directly from active enhancers and therefore mirror enhancer activity in real time. Because enhancers are the master switches that determine spatio-temporal gene expression, eRNAs detect the activation or shutdown of tumor-driving pathways one step earlier than mRNAs or lncRNAs and are less confounded by post-transcriptional regulation. Second, eRNAs have greater tissue specificity and dynamic range. eRNA expression is strictly cell-type- and state-dependent; the same gene can display vastly different enhancer activities across tissues, making eRNAs a high-resolution “tissue fingerprint”. Compared with miRNAs, eRNAs exhibit a wider dynamic range and confer clearer separation between high- and low-risk groups [63]. Third, eRNAs directly linked to actionable epigenetic mechanisms. eRNAs maintain local chromatin accessibility, recruit transcriptional complexes or form 3D chromatin loops, thereby dictating the transcriptional rate of oncogenes or tumour-suppressor genes. This mechanism endows eRNAs with a dual role as both prognostic biomarkers and therapeutic targets (via CRISPRi, antisense oligos or small molecules), whereas traditional miRNAs or lncRNAs often have pleiotropic targets and lower intervention specificity [63].

It is evident that prognostic studies focusing on other RNA species are already abundant, whereas eRNA-based investigations remain scarce and superficial. By constructing the 7-eRNA signature, we provide new insights for clinical prognosis, target discovery and mechanistic exploration in LUAD management.

Also, there are future research prospects of deeper mechanisms such as the interaction between eRNA and genomic alterations and the regulation of chromatin accessibility by eRNA.

## 5. Conclusions

We developed a LUAD prognostic model based on eRNA expression data TCGA. We constructed a 7-eRNA prognostic model using precise screening, and the model was identified to be effective and temperate. The 7-eRNA model robustly stratified LUAD patients into high-risk and low-risk groups in both the train and test sets. Functional analyses revealed distinct enrichment of pathways related to amino acid biosynthesis, ribosome biogenesis, and proteasome activity in high-risk patients. Somatic mutation profiling highlighted TP53 and TTN as frequently mutated genes, while drug sensitivity prediction identified four potential therapeutic agents (including AZD4547 and Nutlin-3a) for high-risk individuals. Collectively, this study constructed a 7-eRNA prognostic model for LUAD, providing a powerful tool for clinical risk assessment and uncovering eRNA-mediated regulatory mechanisms.

In conclusion, the 7-eRNA model is a reliable biomarker for predicting the prognosis of patients with LUAD. Our research provides valuable insights into new potential biomarkers for predicting the prognosis and survival of patients with LUAD. In addition, it is expected to provide opportunities for improving drug treatment.

## Figures and Tables

**Figure 1 biology-14-01431-f001:**
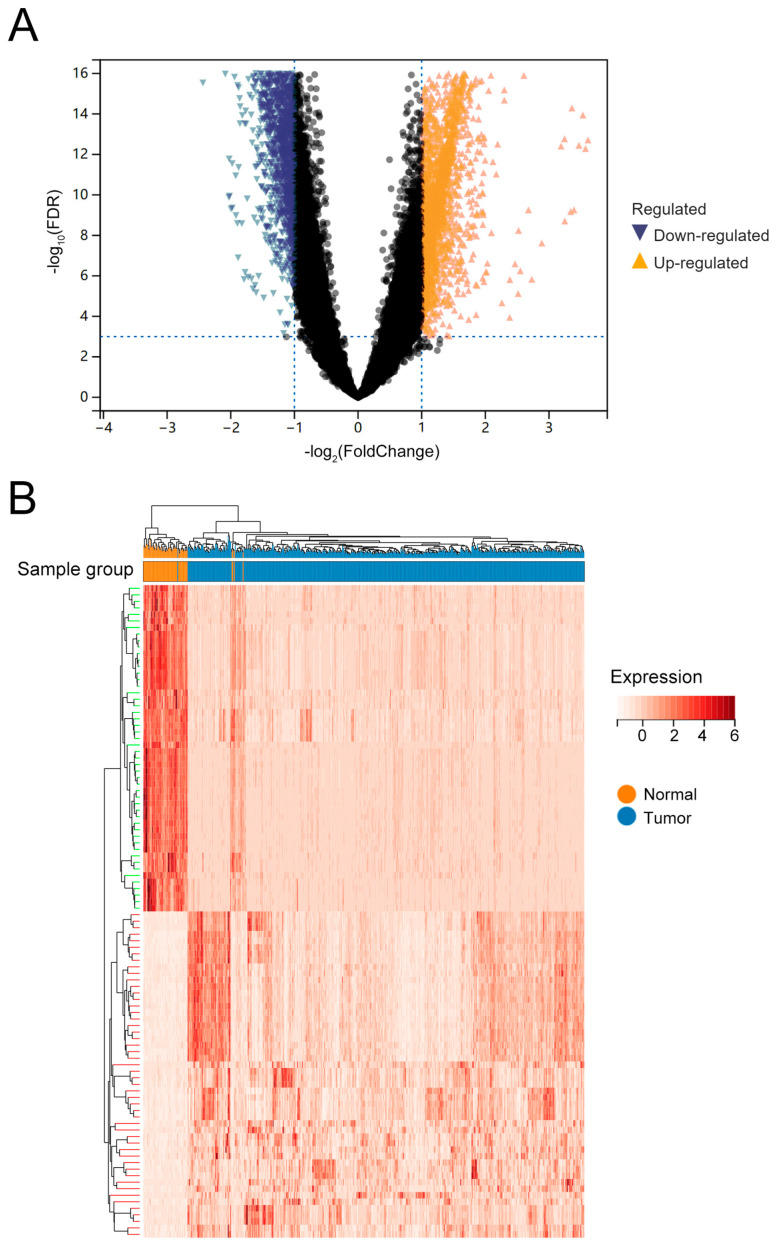
Differentially expressed eRNAs in LUADs. (**A**) Volcano plot showing differential eRNA expression between tumor and normal samples. Yellow indicates upregulated eRNAs, blue indicates downregulated eRNAs, and black indicates eRNAs with insignificant expression changes. (**B**) Heatmap displaying the top 50 differentially expressed eRNAs in tumor and normal samples. Yellow represents the normal group, and blue represents the tumor group.

**Figure 2 biology-14-01431-f002:**
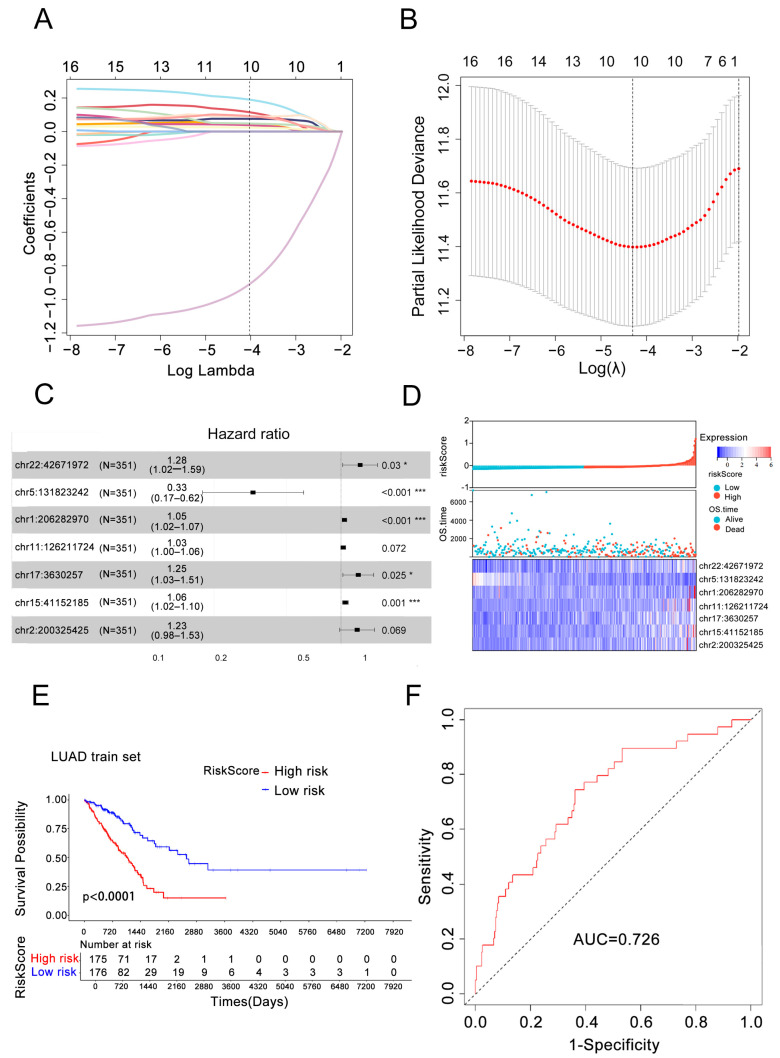
Construction and evaluation of LUAD’s 7-eRNA prediction model. (**A**) LASSO coefficient profile: *X*-axis denotes log(λ) (λ = regularization parameter), *Y*-axis represents feature coefficients, and each line corresponds to an eRNA. As log(λ) increases, eRNA coefficients become sparse and shrink to zero, enabling feature screening. (**B**) LASSO regularization profile: *X*-axis indicates λ values, *Y*-axis represents model performance metrics (e.g., root-mean-square error, cross-validation error). Vertical dashed lines mark the optimal λ range, balancing feature parsimony and predictive performance. (**C**) Multivariate Cox regression identified the optimal 7-eRNA prognostic signature. * *p* < 0.05; *** *p* < 0.001. (**D**) Expression of 7 eRNAs and risk score distribution: Blue indicates the low-risk group (top), surviving patients (middle), and low eRNA expression (bottom); red indicates the high-risk group (top), deceased patients (middle), and high eRNA expression (bottom). (**E**) Kaplan–Meier survival curve: red = high-risk group, blue = low-risk group (log-rank test, *p* < 0.001). “Number at risk” = patients without events; “Cumulative events” = total events in each group. (**F**) ROC curve.

**Figure 3 biology-14-01431-f003:**
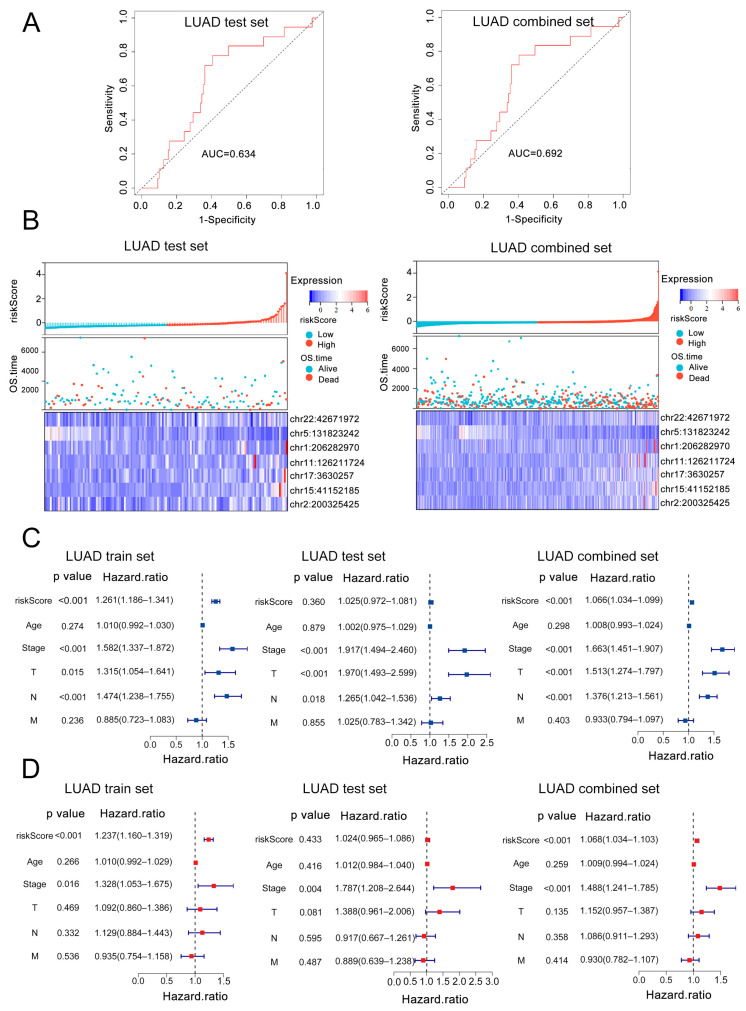
Validation of the 7-eRNA model and robustness assessment across clinical parameters in independent cohorts. (**A**) Risk score heatmaps showing associations between risk scores, OS time, and 7-eRNA expression levels in the test set and combined set. (**B**) ROC curves evaluating the performance of the 7-eRNA model in the test set and combined set. (**C**) Univariate Cox regression analyses of clinical features in the training, test, and combined sets. (**D**) Multivariate Cox regression analyses of clinical features in the training, test, and combined sets. T: Primary tumor size/extent; N: Lymph node metastasis; M: Distant metastasis.

**Figure 4 biology-14-01431-f004:**
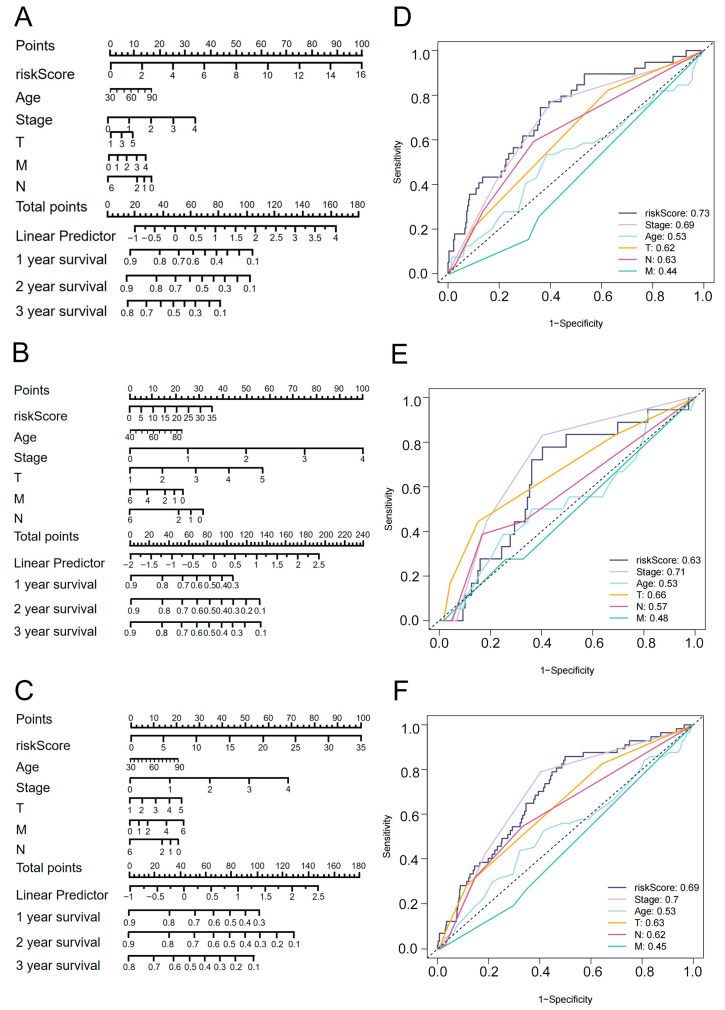
Nomograms and ROC curves for clinical variables. (**A**–**C**) Nomograms predicting OS time in the train set (**A**), test set (**B**), and combined set (**C**). (**D**–**F**) ROC curves comparing the prognostic performance of different clinical variables in the train set (**D**), test set (**E**), and combined set (**F**).

**Figure 5 biology-14-01431-f005:**
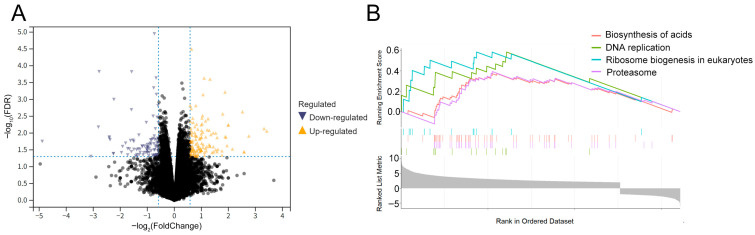
Gene set enrichment analysis. (**A**) Volcano plot showing differentially expressed genes between the high-risk and low-risk groups (FDR < 0.05, |log_2_FC| > 0.4). Yellow indicates upregulation, blue indicates downregulation, and black indicates no significant difference. (**B**) GSEA of the high-risk group.

**Figure 6 biology-14-01431-f006:**
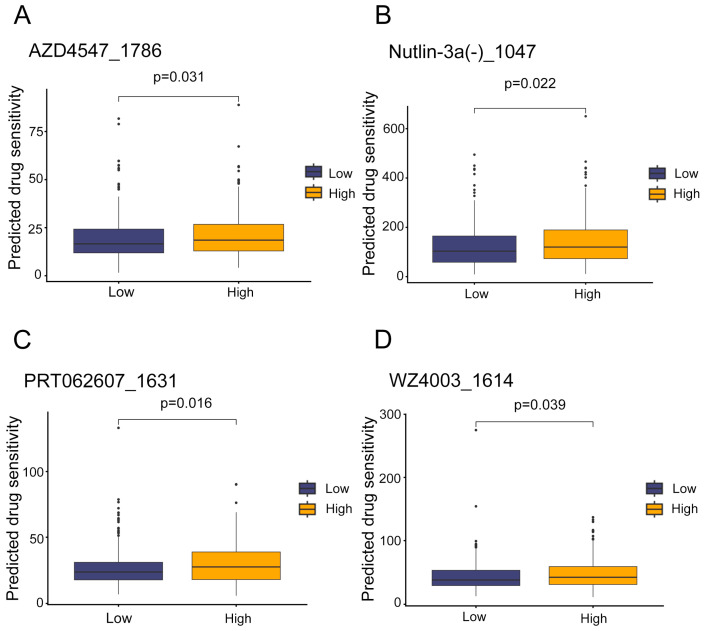
Identification of potential drugs with higher therapeutic sensitivity in high-risk groups (**A**–**D**). The top four drugs with significant differences in predicted sensitivity between the two subgroups are shown. *p*-values were calculated using the Mann–Whitney test.

**Figure 7 biology-14-01431-f007:**
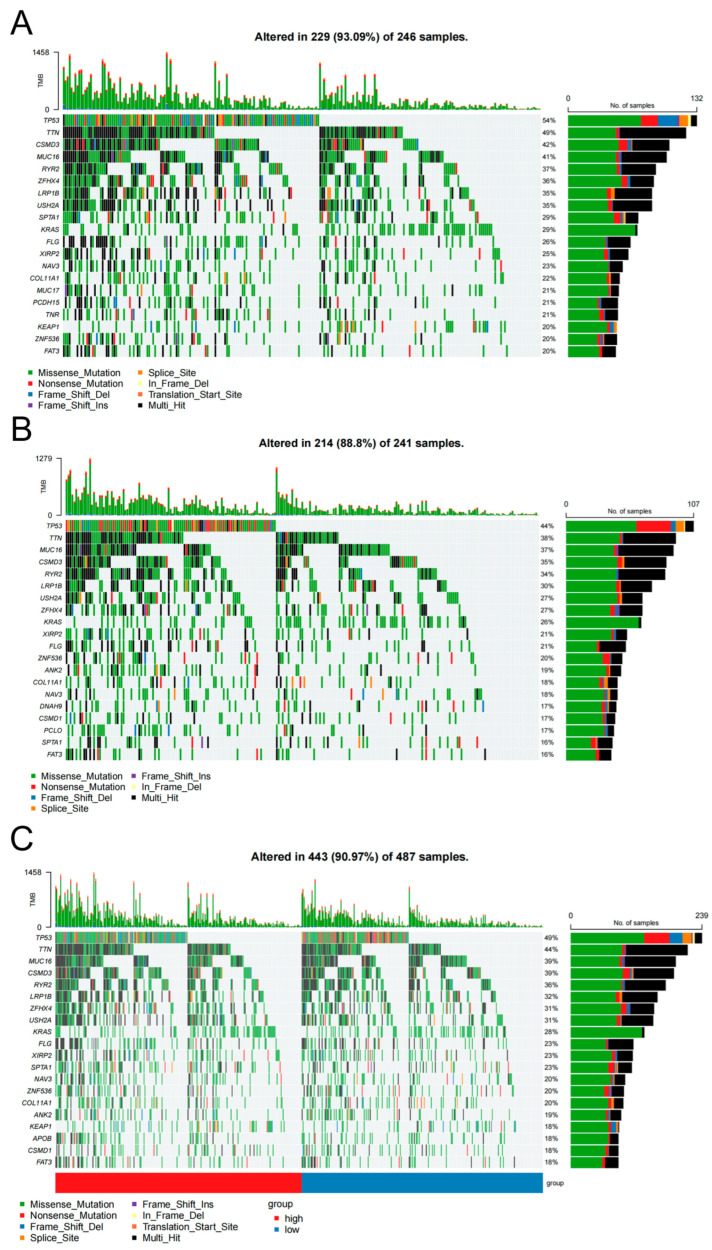
Genomic alteration analysis between high- and low-risk groups. (**A**) Top 20 mutated genes in the high-risk group. (**B**) Top 20 mutated genes in the low-risk group. (**C**) Comparison of mutated gene differences between the two groups; chi-square test was used to assess mutation frequencies of the top 20 genes in both groups.

**Figure 8 biology-14-01431-f008:**
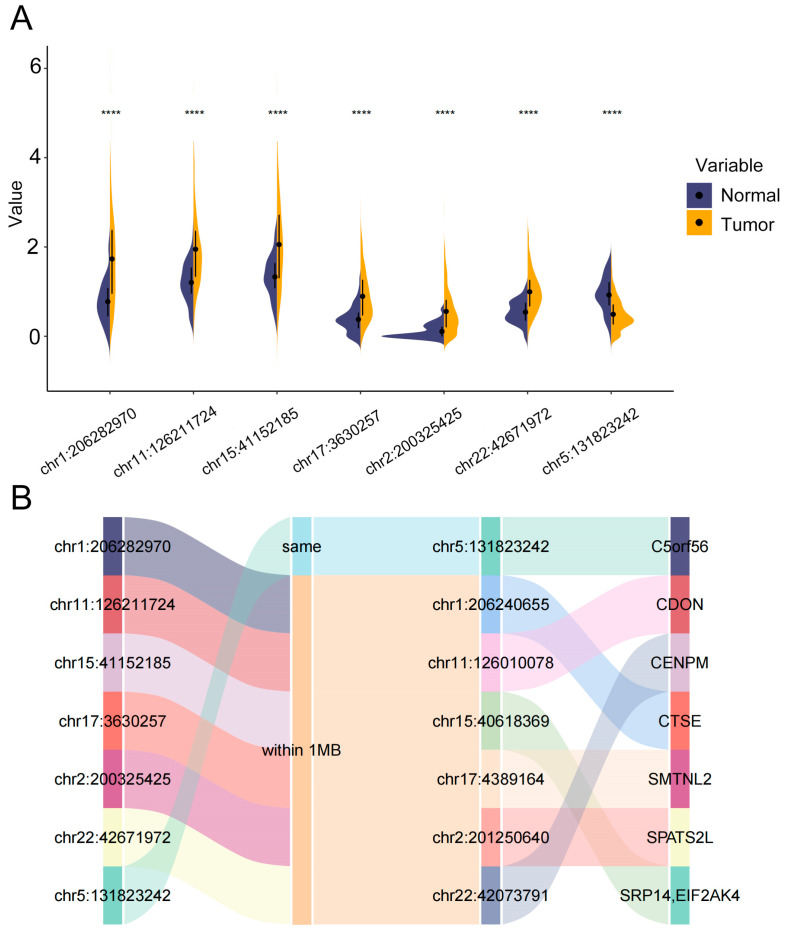
eRNA-gene correspondences from the TCeA portal. (**A**) Differential expression of seven eRNAs in tumor and normal samples (blue = normal samples, yellow = tumor samples; **** *p* < 0.0001). (**B**) Sankey diagram showing potential gene targets of eRNAs.

**Table 1 biology-14-01431-t001:** Information corresponding to numerical encoding during data processing.

Clinical Information	Classification	Numerical Code
Survival status	Alive	0
	Dead	1
Stage	Stage Ⅰ	1
	Stage Ⅱ	2
	Stage Ⅲ	3
	Stage Ⅳ	4
T Classification	T1	1
	T2	2
	T3	3
	T4	4
	TX	5
N Classification	N0	0
	N1	1
	N2	2
	N3	3
M Classification	M0	0
	M1	1
	MX	2

**Table 2 biology-14-01431-t002:** Prognostic eRNAs obtained from multivariable Cox regression analysis.

ID	Coefficient	HR	95% CI	*p* Value
chr1:206282970	0.04578	1.04684	1.02279–1.07146	0.00011
chr2:200325425	0.20554	1.22819	0.98391–1.53313	0.06928
chr5:131823242	−1.11341	0.32844	0.17302–0.62344	0.00066
chr11:126211724	0.02812	1.02852	0.99748–1.06053	0.07210
chr15:41152185	0.05767	1.05936	1.02252–1.09753	0.00141
chr17:3630257	0.22138	1.24780	1.02789–1.51475	0.02521
chr22:42671972	0.24401	1.27636	1.02369–1.59140	0.03016

## Data Availability

RNA-Seq count data of LUAD, cancer patient information and somatic mutation are downloaded from TCGA GDC Data Portal (https://portal.gdc.cancer.gov/ (accessed on 29 September 2024)). The expression levels of eRNA of LUAD are downloaded from TCeA Portal (https://bioinformatics.mdanderson.org/Supplements/Super_Enhancer/TCEA_website/ (accessed on 21 October 2024)). Drug response data were retrieved from GDSC (http://www.cancerrxgene.org/ (accessed on 25 March 2025)).

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
