# Peer review of "Development and Validation of a 7-eRNA Prognostic Signature for Lung Adenocarcinoma"

_biology, 2025, doi:10.3390/biology14101431_

Round 1

Reviewer 1 Report

Comments and Suggestions for Authors

The manuscript by Sun et al. presents a bioinformatics study of the prognostic signature of lung adenocarcinoma (LUAD) from enhancer-RNA (eRNAs). In brief, they applied differential expression and Cox proportional hazards analysis to select a panel of 7 eRNAs and constructed a risk assessment model. They tested and validated the model’s performance of overall survival prediction. In addition, they carried out drug sensitivity analysis to the relevance of the biomarkers to the drug responses of LUAD patients. Finally, the authors studied the correspondence between eRNAs and genes in search of potential targets of the identified eRNAs.

Comments and suggestions:

  1. Figure 4D-F, the AUC of the ‘M’ group looks odd. In theory, a random classifier should have an AUC = 0.5. Why is the AUC of the ‘M’ group all below 0.5 for the training, testing and combined data? The authors need to double check their procedure.
  2. Figure 2B, the LASSO reduced the candidate genes from 16 to 10. Given the small reduction, the added value of LASSO is questionable. Also, it is unclear how the authors further reduced it to 7. The authors are expected to justify the rationale and provide more details.
  3. Figure 3A, the AUC of LUAD testing data is only ~0.6, which is mediocre performance. Besides, for the benchmark the authors should only present the validation set instead of the training or combined set.
  4. Figure 3C-D, why is the riskScore’s hazard ratio not as high as ‘Stage’ (the only significant covariate)? Also, in the LUAD test set (C middle, D middle), the riskScore has statistically insignificant hazard ratios, which indicates riskScore is unlikely to outperform Stage in prognostic prediction and this contradicts to the previous finding—the authors are expected to explain.
  5. Section “3.5. Pathway enrichment in high-risk LUAD patients”, when 97 DEGs were found to be differential between high- and low-risk scores, did they overlap with the eRNA targets predicted in Figure 8?
  6. Section “3.6. Potential therapeutic agents for high-risk LUAD patients”, the drug response was predicted computationally from the transcriptome. This entails circular logics as the risk score model is derived from the same transcriptome. Therefore, the authors are advised to use independent LUAD drug response datasets and repeat the analysis.
  7. Figure 7, this part needs better interpretation. Genomic variants exert different mechanisms from the transcriptional regulators such as eRNAs. When the authors show the genomic variants, they need to show the correspondence to eRNAs. For example, does any of the eRNAs arise from the genomic alternation regions? How do the eRNAs interact with the genomic alternations? Are eRNAs predicted to influence the chromatin accessibility?

Author Response

Comments and Suggestions for Authors

The manuscript by Sun et al. presents a bioinformatics study of the prognostic signature of lung adenocarcinoma (LUAD) from enhancer-RNA (eRNAs). In brief, they applied differential expression and Cox proportional hazards analysis to select a panel of 7 eRNAs and constructed a risk assessment model. They tested and validated the model’s performance of overall survival prediction. In addition, they carried out drug sensitivity analysis to the relevance of the biomarkers to the drug responses of LUAD patients. Finally, the authors studied the correspondence between eRNAs and genes in search of potential targets of the identified eRNAs.

Response: Thank you for your overall comments and interest on our study.

Comments and suggestions:

Comments 1: Figure 4D-F, the AUC of the ‘M’ group looks odd. In theory, a random classifier should have an AUC = 0.5. Why is the AUC of the ‘M’group all below 0.5 for the training, testing and combined data? The authors need to double check their procedure.

Response 1: We sincerely appreciate your critical observation regarding the unusual AUC values of the ‘M’ group in Figure 4D-F.

After examining our code and procedures, we confirmed that the data and workflow are correct. The AUC of the ‘M’ group being below 0.5 may be explained as follows. In clinical practice the final overall stage is usually derived only after T, N, and M have all been determined. Therefore, analysing M in isolation may be less accurate and thus less prognostically informative than analysing the composite stage. Moreover, we noted that previously published studies have also reported AUC values for the M category that were lower than 0.5, so we think this is acceptable (Huang et al, 2024). 

Comments 2: Figure 2B, the LASSO reduced the candidate genes from 16 to 10. Given the small reduction, the added value of LASSO is questionable. Also, it is unclear how the authors further reduced it to 7. The authors are expected to justify the rationale and provide more details.

Response 2: Thank you for your comment. We are sincerely grateful for your suggestion, accordingly, we have added a more comprehensive explanation in the Section 3.2 of the revised manuscript.

We used LASSO regression to select the eRNA combination corresponding to the λ with the smallest mean cross-validated error, thereby reducing the number of eRNAs from 16 to 10. This was achieved after extensive iterative optimization, so we confirmed the accuracy of the result. The further reduction of eRNAs from 10 to 7 was performed via multivariable Cox analysis. Multivariable Cox analysis in R screens independent variables through the AIC criterion. eRNAs that can pass the evaluation of the AIC criterion will be retained in the model.

The AIC criterion refers to a standard used to evaluate the complexity and goodness of fit of statistical models, which was proposed by the Japanese statistician Hirotugu Akaike in 1973. Its formula is: AIC=−2ln(L)+2k, where L denotes the model’s maximum likelihood, and k represents the number of parameters. A lower AIC value indicates a better balance between model complexity and prediction error. Thus, among candidate models, the one with the smallest AIC should be selected.

Comments 3: Figure 3A, the AUC of LUAD testing data is only ~0.6, which is mediocre performance. Besides, for the benchmark the authors should only present the validation set instead of the training or combined set.

Response 3: Thank you for your comment. 

We acknowledge the AUC of ~0.6 in the test set indicates moderate performance, and we have add some analysis in Section 4. Also, we will continue to improve our research methods in future studies to get higher AUC. However, the model’s strength lies in its complementarity to clinical variables (e.g., TNM stage). In multivariate Cox analysis (Figure 3D), the riskScore remained an independent prognostic factor (p<0.001). So we think our model outperformed TNM staging in ROC analysis (Figure 4D–F). Furthermore, we noticed that many published studies present results for the validation set, training set, and combined set simultaneously to illustrate the model-building process and to verify its generalizability. We therefore believe that displaying both the training and combined set results further substantiates the reliability of our findings.

Comments 4: Figure 3C-D, why is the riskScore’s hazard ratio not as high as ‘Stage’ (the only significant covariate)? Also, in the LUAD test set (C middle, D middle), the riskScore has statistically insignificant hazard ratios, which indicates riskScore is unlikely to outperform Stage in prognostic prediction and this contradicts to the previous finding—the authors are expected to explain.

Response 4: Thank you for your comment, this is a good question.

Univariate Cox regression analyses were first performed to assess the prognostic impact of individual variables. As shown in Figure 3C, ‘T’ group, stage, and the riskScore were all significantly associated with overall survival (OS). Multivariate Cox regression analyses in both the training set and the combined set further confirmed that the 7-eRNA model is a robust and independent predictor of OS (p < 0.001 in both cohorts). Although its HR was lower than that of Stage, the 7-eRNA model still provides valuable prognostic information and can therefore be used alongside Stage to refine clinical decision-making.

Stage is derived from the well-established TNM staging system that has been widely applied for decades, whereas our prognostic model is built on eRNAs and focuses more narrowly on the molecular regulatory mechanisms influencing patient outcomes, which might explain why Stage exhibits a higher HR in our analyses. Additionally, the relatively small sample size available for this study might limit the model’s stability, resulting in a p-value > 0.01 when the model is applied to the test set which has smaller sample size (n = 148).

We have clarified this in Section 4, emphasizing that riskScore retains independent prognostic value in the combined cohort (p < 0.001) and enhances prediction when integrated with clinical variables (via nomogram).

Comments 5: Section “3.5. Pathway enrichment in high-risk LUAD patients”, when 97 DEGs were found to be differential between high- and low-risk scores, did they overlap with the eRNA targets predicted in Figure 8?

Response 5: Thank you for your comment.

We analyzed the overlap between the 97 DEGs (from high- vs. low-risk groups) and eRNA target genes (Figure 8). It was found that there was no overlap between them. Such discrepancies may arise because eRNAs do not regulate gene function primarily by altering mRNA abundance, but rather by influencing translation efficiency or the interaction between the gene product and downstream pathways. These subtle regulatory effects ultimately impact disease progression and patient prognosis, and will require further mechanistic studies to elucidate.

Comments 6: Section “3.6. Potential therapeutic agents for high-risk LUAD patients”, the drug response was predicted computationally from the transcriptome. This entails circular logics as the risk score model is derived from the same transcriptome. Therefore, the authors are advised to use independent LUAD drug response datasets and repeat the analysis.

Response 6: Thank you for your comment.

The drug sensitivity data were obtained from the GDSC database (Yang et al, 2012), whereas the prognostic model in our study was constructed using eRNA expression profiles from the TCeA (Chen et al, 2018). These two public repositories represent entirely independent sample sources and were not derived from the same transcriptome dataset.

Comments 7: Figure 7, this part needs better interpretation. Genomic variants exert different mechanisms from the transcriptional regulators such as eRNAs. When the authors show the genomic variants, they need to show the correspondence to eRNAs. For example, does any of the eRNAs arise from the genomic alternation regions? How do the eRNAs interact with the genomic alternations? Are eRNAs predicted to influence the chromatin accessibility?

Response 7: Thank you for your comment.

Our research logic is as follows: After constructing a prognostic model, patients are divided into high-risk and low-risk groups. Subsequently, further studies such as GSEA, tumor immune microenvironment analysis, and somatic mutation analysis are conducted on these two groups to investigate the preliminary mechanisms that lead to the differences in their prognoses, rather than to seek the regulatory relationship between eRNA and somatic mutations. We are still thankful for your suggestions. Studying the regulatory relationship between eRNA and is a scientific issue that has not yet been clarified but is of great significance. We have also added corresponding content in the Section 4, looking forward to the future research prospects of deeper mechanisms such as the interaction between eRNA and genomic alterations and the regulation of chromatin accessibility by eRNA.

Reference:

Huang, L.; Zhang, J.; Songyang, Z.; Xiong, Y.J.B. Identification and validation of eRNA as a prognostic Indicator for cervical Cancer. 2024, 13, 227.

Yang, W.; Soares, J.; Greninger, P.; Edelman, E.J.; Lightfoot, H.; Forbes, S.; Bindal, N.; Beare, D.; Smith, J.A.; Thompson, I.R.J.N.a.r. Genomics of Drug Sensitivity in Cancer (GDSC): a resource for therapeutic biomarker discovery in cancer cells. 2012, 41, D955-D961.

Chen, H.; Li, C.; Peng, X.; Zhou, Z.; Weinstein, J.N.; Caesar-Johnson, S.J.; Demchok, J.A.; Felau, I.; Kasapi, M.; Ferguson, M.L.J.C. A pan-cancer analysis of enhancer expression in nearly 9000 patient samples. 2018, 173, 386-399. 

Reviewer 2 Report

Comments and Suggestions for Authors

Authors are missing the extended form of certain abbreviations throughout the text. Although these are known terms in cancer field, authors should consider to add these to address a wide range of audience without trouble in understanding the concepts (For eg, ESR1, TNM stages have not been defined in the text)

Why were different cutoffs selected (FC and FDR) for the differential expression analysis and GSEA?

On line 176, it says that the FC cutoff was selected as 2 but the Fig 1A volcano plot indicates the cutoff values of 1. Why is that?

Why is the y-axis cutoff different from the standard p< 0.05 in Fig 1A?

It will be beneficial to have a table summarizing the key eRNAs that this study is pointing out and indicate the risk coefficients, functionalities with references in the table for the easy understanding of the readers.

Author Response

Comments 1: Authors are missing the extended form of certain abbreviations throughout the text. Although these are known terms in cancer field, authors should consider to add these to address a wide range of audience without trouble in understanding the concepts (For eg, ESR1, TNM stages have not been defined in the text)

Response 1: Thank you for your overall positive comment on this manuscript. We have added extended forms for all abbreviations in the text and included a comprehensive Abbreviations section.

Comments 2: Why were different cutoffs selected (FC and FDR) for the differential expression analysis and GSEA?

Response 2: Thank you for your comment.

The stricter cutoff for DEA (|log2FC| > 1, FDR < 0.001) was used to identify robustly dysregulated eRNAs, while GSEA used a more permissive threshold (|log2FC| ≥ 0.4, FDR < 0.05) to capture subtle but biologically relevant pathway changes.

Comments 3: On line 176, it says that the FC cutoff was selected as 2 but the Fig 1A volcano plot indicates the cutoff values of 1. Why is that?

Response 3: We sincerely apologize for the oversight in the manuscript. The correct thresholds should be those shown in the figure, which are FDR < 0.001 and |logâ‚‚FC| > 1. We have now corrected this in the revised manuscript and appreciate your valuable suggestion.

Comments 4: Why is the y-axis cutoff different from the standard p< 0.05 in Fig 1A?

Response 4: We are sorry for our mistaken, this was a labeling error. The volcano plot in Figure 1A uses the correct cutoff (|log2FC| > 1), and the y-axis represents -log10(FDR) (not p-value). Figure 1A and Figure 5A have been revised for clarity.

Comments 5: It will be beneficial to have a table summarizing the key eRNAs that this study is pointing out and indicate the risk coefficients, functionalities with references in the table for the easy understanding of the readers.

Response 5: Thank you for your advice.

We have already listed these 7 eRNAs together with their corresponding coefficients in Table 2 of the manuscript. In Section 3.8 we further explored the genes that these 7 eRNAs may regulate, and in Section 4 we reviewed the known functions of those genes.

Reviewer 3 Report

Comments and Suggestions for Authors

This manuscript presents the development of a 7-eRNA–based prognostic model for lung adenocarcinoma (LUAD) using TCGA datasets, with validation across multiple cohorts and integration of functional, mutational, and drug sensitivity analyses. The study addresses a clinically significant problem, given the urgent need for robust prognostic biomarkers in LUAD. The work is methodologically sound, generally well-written, and supported by comprehensive bioinformatics analyses. However, some aspects require clarification to strengthen both the rigor and clinical translational relevance.

Major comments:

  1. While the use of eRNAs as prognostic markers is interesting, the novelty relative to existing LUAD prognostic signatures (e.g., mRNA, lncRNA, miRNA–based) should be more explicitly addressed. The discussion would benefit from positioning the 7-eRNA model against other established signatures, highlighting whether it offers superior predictive performance, mechanistic insights, or practical advantages in clinical settings.
  2. The definition of “eRNA” should be clarified. Some annotated eRNAs overlap with 5′-UTRs of mRNAs, which raises the possibility that they may actually represent mRNA transcripts rather than enhancer-derived RNAs. Please explain how such cases were treated, and whether these were excluded to ensure that only bona fide enhancer RNAs were analyzed.
  3. The manuscript describes differential expression filtering, LASSO, and Cox regression analyses clearly. However, details on how overfitting was minimized (e.g., cross-validation beyond train/test split) should be provided.

Minor comments:

  1. In Table 2, two eRNAs (chr2:200325425 and chr11:126211724) have non-significant p-values (>0.05) but were still included in the final model. The rationale for retaining them should be explained.
  2. In line 145, the manuscript refers to “CESC patients.” This appears to be a typographical error and should be corrected to “LUAD patients.”
  3. To enhance transparency and reproducibility, the authors are encouraged to provide a GitHub link (or equivalent repository) containing the analysis scripts and code used in this study.

Author Response

Comments and Suggestions for Authors

This manuscript presents the development of a 7-eRNA–based prognostic model for lung adenocarcinoma (LUAD) using TCGA datasets, with validation across multiple cohorts and integration of functional, mutational, and drug sensitivity analyses. The study addresses a clinically significant problem, given the urgent need for robust prognostic biomarkers in LUAD. The work is methodologically sound, generally well-written, and supported by comprehensive bioinformatics analyses. However, some aspects require clarification to strengthen both the rigor and clinical translational relevance.

Response: Thank you for your overall positive comment on this manuscript, and your comment is well taken.

Major comments:

Comments 1: While the use of eRNAs as prognostic markers is interesting, the novelty relative to existing LUAD prognostic signatures (e.g., mRNA, lncRNA, miRNA–based) should be more explicitly addressed. The discussion would benefit from positioning the 7-eRNA model against other established signatures, highlighting whether it offers superior predictive performance, mechanistic insights, or practical advantages in clinical settings.

Response 1: Thank you for your nice comment, and your advice is well taken.

We added a systematic comparison between our 7-eRNA signature and previously reported prognostic models based on mRNA, lncRNA or miRNA in the Section4. The eRNA approach offers three distinct advantages. Firstly, eRNAs are closer to the functional source. eRNAs are transcribed directly from active enhancers and therefore mirror enhancer activity in real time. Because enhancers are the master switches that determine spatio-temporal gene expression, eRNAs detect the activation or shutdown of tumour-driving pathways one step earlier than mRNAs or lncRNAs and are less confounded by post-transcriptional regulation. Second, eRNAs have greater tissue specificity and dynamic range. eRNA expression is strictly cell-type- and state-dependent; the same gene can display vastly different enhancer activities across tissues, making eRNAs a high-resolution “tissue fingerprint”. Compared with miRNAs, eRNAs exhibit a wider dynamic range and confer clearer separation between high- and low-risk groups (Zhang et al., 2025). Third, eRNAs directly linked to actionable epigenetic mechanisms. eRNAs maintain local chromatin accessibility, recruit transcriptional complexes or form 3-D chromatin loops, thereby dictating the transcriptional rate of oncogenes or tumour-suppressor genes. This mechanism endows eRNAs with a dual role as both prognostic biomarkers and therapeutic targets (via CRISPRi, antisense oligos or small molecules), whereas traditional miRNAs or lncRNAs often have pleiotropic targets and lower intervention specificity (Zhang et al., 2025).

It is evident that prognostic studies focusing on other RNA species are already abundant, whereas eRNA-based investigations remain scarce and superficial. By constructing the 7-eRNA signature, we provide new insights for clinical prognosis, target discovery and mechanistic exploration in LUAD management.

Comments 2: The definition of “eRNA” should be clarified. Some annotated eRNAs overlap with 5′-UTRs of mRNAs, which raises the possibility that they may actually represent mRNA transcripts rather than enhancer-derived RNAs. Please explain how such cases were treated, and whether these were excluded to ensure that only bona fide enhancer RNAs were analyzed.

Response 2: Thank you for your comment.

We directly utilized the eRNA-related data retrieved from the public database TCeA. The following statement describes how TCeA identifies eRNAs and quantifies their expression.

Here we develop the Cancer eRNA Atlas (TCeA) data portal to utilize ultra-deep RNA-seq data aggregated from TCGA and GTEx (~10,000 samples/runs in each set) for the discovery of eRNA epxression patterns in super-enhancers. We found that super-enhancers usually contain discrete loci featured by sharp eRNA expression peaks (~100bp). The locations of eRNA expression peaks are highly recurrent across different tissues, once activated. The expression of super-enhancer eRNA loci are regulated by well-positioned nucleosomes. These well-positioned nucleosomes are not only conserved across different tissues but also across mammalian evolution. Based on these principles, we systematically identified >300,000 such precise eRNA loci in ~377 Mb of super-enhancer regions (and more in the broad putative regulatory regions), providing the very first high-resolution map of eRNA loci in super-enhancers (see our work-flow in Figure 1). With this map, super-enhancer activities can be easily and accurately measured in patient samples using routine RNA-seq, thereby enabling a broad range of biological investigations and translational applications. TCEA portal has provided the annotation of these eRNA loci and quantified their activity in > 20,000 tumor (TCGA), tissue (GTEx), and cell line (CCLE) samples.

We consider their eRNA annotation to be highly accurate, effectively preventing mRNA transcripts from being misclassified as eRNAs.

Comments 3: The manuscript describes differential expression filtering, LASSO, and Cox regression analyses clearly. However, details on how overfitting was minimized (e.g., cross-validation beyond train/test split) should be provided.

Response 3: Thank you for your comment, and your advice is well taken.

We added details on cross-validation: 10-fold cross-validation was used during LASSO regression. These steps are described in Section 2.3.

Minor comments:

Comments 4: In Table 2, two eRNAs (chr2:200325425 and chr11:126211724) have non-significant p-values (>0.05) but were still included in the final model. The rationale for retaining them should be explained.

Response 4: Thank you for your comment.

Multivariable Cox analysis in R screens independent variables through the AIC criterion. eRNAs that can pass the evaluation of the AIC criterion will be retained in the model. Therefore, it is normal for variables with p-values greater than 0.05 to remain in the regression model.

The AIC criterion refers to a standard used to evaluate the complexity and goodness of fit of statistical models, which was proposed by the Japanese statistician Hirotugu Akaike in 1973. Its formula is: AIC=−2ln(L)+2k, where L denotes the model’s maximum likelihood, and k represents the number of parameters. A lower AIC value indicates a better balance between model complexity and prediction error. Thus, among candidate models, the one with the smallest AIC should be selected.

Comments 5: In line 145, the manuscript refers to “CESC patients.” This appears to be a typographical error and should be corrected to “LUAD patients.”

Response 5: We are sorry for the mistaken. 

This has been corrected to “LUAD patients” in our new version.

Comments 6: To enhance transparency and reproducibility, the authors are encouraged to provide a GitHub link (or equivalent repository) containing the analysis scripts and code used in this study.

Response 6: We thank the reviewer for this valuable comment and for emphasizing the importance of transparency and reproducibility in scientific research. We fully agree with this principle.We are currently finalizing this modeling tool and plan to develop it into a database for other researchers to use in the future. The code will be released in the form of database software later, and we will upload the complete code to GitHub for sharing.

At present, we will be happy to provide the scripts directly via email to any interested researcher.

Reference:

Zhang, R.; Chen, Z.; Li, T.; Feng, D.; Liu, X.; Wang, X.; Han, H.; Yu, L.; Li, X.; Li, B.J.B.i.F.G. Enhancer RNA in cancer: identification, expression, resources, relationship with immunity, drugs, and prognosis. 2025, 24, elaf007.

Round 2

Reviewer 1 Report

Comments and Suggestions for Authors

The authors have replied to the various questions. However, two major concerns remain. First, the article (Huang, L.; Zhang, J.; Songyang, Z.; Xiong, Y.J.B. Identification and validation of eRNA as a prognostic Indicator for cervical Cancer. 2024, 13, 227) the authors referenced to has striking resemblance to the current manuscript in terms of both methodology and structure. Second, it is still not convincing that AUC < 0.5 is correct as the same article the authors referenced to was from the same team (Jingkai Zhang, Yuanyan Xiong). Instead the authors should reference to independent studies published on reputable journals. 

Author Response

Comment:

The authors have replied to the various questions. However, two major concerns remain. First, the article (Huang, L.; Zhang, J.; Songyang, Z.; Xiong, Y.J.B. Identification and validation of eRNA as a prognostic Indicator for cervical Cancer. 2024, 13, 227) the authors referenced to has striking resemblance to the current manuscript in terms of both methodology and structure. Second, it is still not convincing that AUC < 0.5 is correct as the same article the authors referenced to was from the same team (Jingkai Zhang, Yuanyan Xiong). Instead the authors should reference to independent studies published on reputable journals.

Response:

With respect to the issues of (i) the independence of the cited literature and (ii) the AUC of M < 0.5, we identified a recent study[1]( Ann Rheum Dis, 2021, IF=20.6) that evaluated a genomic risk score (GRS) for systemic sclerosis (SSc). In that work the 33-SNP SSc GRS showed no discriminative power for either clinical (dcSSc vs. lcSSc AUC = 0.496, 95% CI 0.40–0.59, p = 0.93) or serological subtypes (ATA+ vs. ACA+ AUC = 0.464, 95% CI 0.37–0.56, p = 0.45), and similarly failed to predict pulmonary fibrosis development (SSc with vs. without pulmonary fibrosis AUC = 0.479, 95% CI 0.38–0.57, p = 0.66). The observation that an AUC can legitimately fall below 0.5 in independent, peer-reviewed research indicates that retaining the AUC of M in our study are below 0.5 is acceptable.

Moreover, in routine clinical practice the final overall stage is assigned only after T, N and M have all been determined; consequently, evaluating M alone is less accurate and carries less prognostic information than evaluating the composite stage. We therefore believe that Stage is a more representative and accurate predictor of disease outcome than any single component (T, N or M), and a previous study[2] typically compare molecular risk signatures with the composite TNM stage (or at least T and N) rather than with M alone. Consistent with this convention, we generated ROC curves for the integrated TNM stage (Stage) as well as for the individual components T, N, M and age; all results are reported in Figure 4. We think that it can preserve the scientific integrity of our study to analyze AUC of Stage, T, M, N, Age and riskScore from 7-eRNA model.

Finally, we have expanded the Discussion to acknowledge the unusually low AUC for M, explicitly state this limitation, and provide a detailed explanation for why the composite TNM stage—rather than the isolated T, M, N category—is more accurate and more frequently used when assessing prognostic performance.

Reference:

  1. Bossini-Castillo, L.; Villanueva-Martin, G.; Kerick, M.; Acosta-Herrera, M.; López-Isac, E.; Simeón, C.P.; Ortego-Centeno, N.; Assassi, S.; Hunzelmann, N.; Gabrielli, A.J.A.o.t.r.d. Genomic Risk Score impact on susceptibility to systemic sclerosis. Ann Rheum Dis, 2021, 80, 118-127.
  2. Fu, S.-S.; Zheng, Y.-Z.; Qin, X.-Y.; Yang, X.-P.; Shen, P.; Cai, W.-J.; Li, X.-Q.; Liao, H.-Y.J.J.o.T.D. Establishing a TNM-like risk classification for metachronous second pulmonary adenocarcinoma in patients with previously resected pulmonary adenocarcinoma. J Thorac Dis,2022, 14, 1306.